# Knockdown resistance (*kdr*) associated organochlorine resistance in mosquito-borne diseases (*Culex quinquefasciatus*): Systematic study of reviews and meta-analysis

**Ebrahim Abbasi**[1,2]*, **Salman Daliri**[3]

**1** Research Center for Health Sciences, Institute of Health, Shiraz University of Medical Sciences, Shiraz, Iran, **2** Department of Biology and Control of Disease Vectors, School of Health, Shiraz University of Medical Sciences, Shiraz, Iran, **3** Health Deputy, Shiraz University of Medical Sciences, Shiraz, Iran

* abbasie.ebrahim@gmail.com, e_abbasie@sums.ac.ir

## Abstract

### Introduction

*Culex quinquefasciatus* is one of the most important carriers of human pathogens. Using the insecticides is one of the most important methods of combating this vector. But the genetic resistance created in *Culex quinquefasciatus* led to disruption in the fight against this pest. Consequently, it is necessary to know the level of resistance to fight this vector. Based on this, the present study was conducted to investigate the prevalence of *kdr* resistance in *Culex quinquefasciatus* against organochlorine insecticides in the world.

### Methods

This study was conducted by systematic review, and meta-analysis on the prevalence of *kdr* resistance and mortality rate in *Culex quinquefasciatus* against organochlorine insecticides in the world. All pertinent articles were extracted and analyzed in accordance with this information during an unrestricted search of the scientific databases Web of Science, PubMed, Scopus, biooan.org, Embase, ProQuest, and Google Scholar until the end of November 2023. Statistical analysis of data was done using fixed and random effects model in meta-analysis, $I^2$ index, Cochran's test, and meta-regression by STATA version 17 software.

### Results

Seventy articles were included in the meta-analysis process. Based on the findings, the prevalence of *Kdr* in *Culex quinquefasciatus* against organochlorine insecticide was estimated at 63.1%. Moreover, the mortality rate against the insecticide deltamethrin was 46%, DDT 18.5%, permethrin 42.6%, malathion 54.4% and lambdacyhalothrin 53%.

### Conclusion

More than half of Cx. quinquefasciatus had *Kdr*. This vector was relatively resistant to DDT and permethrin insecticides and sensitive to malathion, deltamethrin and

**Data Availability Statement:** All relevant data are within the paper and its Supporting Information files.

**Funding:** The author(s) received no specific funding for this work.

**Competing interests:** The authors have declared that no competing interests exist.

lambdacyhalothrin. In order to prevent the development of resistance to alternative insecticides, it is consequently critical to combat this vector with efficacious insecticides.

## Author summary

Using the insecticides is one of the most important methods of combating this vector. Based on this, the present study was conducted to investigate the prevalence of kdr resistance in *Culex quinquefasciatus* against organochlorine insecticides in the world. This study was conducted by systematic review, and meta-analysis on the prevalence of kdr resistance and mortality rate in *Culex quinquefasciatus* against organochlorine insecticides in the world. Based on the findings, the prevalence of Kdr in *Culex quinquefasciatus* against organochlorine insecticide was estimated at 63.1%. Moreover, the mortality rate against the insecticide deltamethrin was 46%, DDT 18.5%, permethrin 42.6%, malathion 54.4% and lambdacyhalothrin 53%. More than half of *Cx. quinquefasciatus* had Kdr. This vector was relatively resistant to DDT and permethrin insecticides and sensitive to malathion, deltamethrin and lambdacyhalothrin.

## Introduction

Mosquitoes carry various types of diseases, including malaria, dengue fever, West Nile fever, yellow fever, filariasis, Japanese encephalitis, and Zika, especially in tropical and subtropical regions of the world. The genus Culex is one of the types of mosquitoes which has 550 species. *Culex quinquefasciatus* species, *Cx. fuscocephala*, *Cx. pseudovishnui*, *Cx. gelides*, *Cx. Tritaenorhynchus*, and *Cx. vishnui* are carriers of diseases, including Japanese encephalitis, Bancroftian filariasis, West Nile virus, and St. Louis encephalitis virus [1]. *Culex quinquefasciatus* is the major vector of Bancroftian filariasis in Asia, South Africa and India [1]. Considering the endemicity of Bancroftian filariasis in India, about six hundred million people are exposed to this disease [2].

The utilization of synthetic insecticides for vector control has been the cornerstone approach in the realm of vector-borne disease management for numerous decades. Consequently, people used insecticides in the form of indoor residual sprays (IRS), insecticide-impregnated nets (ITNs), and LLINs to control and fight these vectors and prevent the spread of diseases. But, the widespread and indiscriminate use of chemical insecticides over time led to resistance in the vector population [3,4].

Resistance developed in insects against insecticides is carried out via four mechanisms, which include 1) modifying or changing behavior 2) reducing the penetration of insecticides into the body, by increasing the thickness of the cuticle or changing the composition of the cuticle 3) metabolic detoxification of insecticide which is done by strengthening or changing key enzymes. 4- Mutation in the target location and reducing their sensitivity [5,6]. Mutations in the target site and metabolic detoxification by enzymes are among the most important and widespread mechanisms of resistance to chemical insecticides [7]. Knockdown resistance (*kdr*) is a prevalent form of resistance observed in insects, wherein a mutation in the voltage-sensitive sodium channel gene (Vssc) induces insensitivity to the target site [6]. Organophosphates, organochlorines, pyrethroids, and carbamates are four main groups of insecticides to fight against vectors [8]. Organochlorines are widely used to control vectors, including *Culex quinquefasciatus*. The insecticides in question exert an impact on both the central and

peripheral nervous systems of insects. By manipulating the sodium channel, which increases its susceptibility to depolarization, and by impeding the inactivation processes, these chemicals ultimately result in the immobilization and demise of the vectors [9,10]. This type of resistance leads to the neutralization of DDT and Deltamethrin to fight against vectors and reduce their nervous sensitivity to these insecticides [11].

*Cx. quinquefasciatus* resistance against insecticides was reported from different regions of the world and is increasing [12,13]. Different studies showed the presence of *kdr* resistance and reduced effect of organochlorine and organophosphorus insecticide in *Cx. quinquefasciatus* reported, but the prevalence of these resistances was different in different regions of the world [14,15]. Based on this; to fight this vector, it is necessary to know the resistance and choose the right insecticide. Considering the spread of this vector and the diseases caused by it in the world, this study was carried out with the aim of the prevalence of *Kdr* and the mortality rate against organochlorine insecticides in the world by systematic review, and meta-analysis.

## Materials and methods

This systematic review and meta-analysis study was conducted on the prevalence of *kdr* in *Cx. quinquefasciatus* and resistance to organochlorine insecticides based on the guidelines of Preferred Reporting Items for Systematic Reviews and Meta-Analyses (PRISMA) [16]. This research was registered in the International Prospective Register of Systematic Review (PROSPERO) with the code CRD42021231605.

### Search strategy

By searching in scientific databases Web of Science, PubMed, Embase, ProQuest, biooan.org, Scopus, and Google Scholar and using the keywords knockdown resistance, Resistance, *kdr*, Organochlorine Insecticide, Insecticide, Chlorinated Insecticide, Chlorophenyl, DDT, malathion, dichloroethane, para chlorophenyl, dichlorodiphenyldichloroethane, dieldrin, deltamethrin, permethrin, *Culex*, *Culex quinquefasciatus*, all related articles were extracted without time limit until the end of 2023. Searching for keywords was individually done and in combination using OR, AND, and NOT operators in the title, abstract, and full text of the articles.

### Inclusion and exclusion criteria

The inclusion criteria included 1- English language articles that were conducted on *Cx. quinquefasciatus*, 2- The prevalence of resistance or mortality in exposure to organochlorine insecticides was reported or estimated in them, 3- *kdr* resistance was investigated in them. 4- and they had good quality. The following were excluded criteria: research focused on other insect species; failure to investigate *kdr* resistance or mortality; qualitative research methods not meeting the desired standards; case reports or narratives; letters to the editor; or articles that were merely reviews or narratives.

### Quality assessment

Using STROBE (Strengthening the Reporting of Observational Studies in Epidemiology) checklist, the quality of articles was measured. This checklist in the field of observing the principles of research writing, and implementation has the areas of title, project implementation, research findings, limitations, and conclusions. Each part of this checklist has subgroups and points are given based on their importance. The maximum score that can be obtained is 33 and the obtained score is more than 20 is acceptable [17].

## Data extraction

At first, considering the inclusion and exclusion criteria, the title and abstract of all articles were reviewed by two researchers independently. If the articles were related, full text was checked, and if they were unrelated, they were excluded from the study. In the cases where there was a difference of opinion between two researchers, the article was refereed by a third person. Data extraction was done using a pre-prepared checklist that included the first author's name, year of study, study location, sample size, insecticide type, *kdr* resistance prevalence, and mortality rate.

## Selection of studies

The number of 16,852 articles extracted from scientific databases were entered into Endnote software. At first, duplicate articles were reviewed, and 7654 articles were removed due to duplicates. In the next step, after reviewing title, abstract, and the full text of the articles, 9185 articles were removed due to being unrelated or not investigating *kdr* resistance, resistance to organochlorine insecticide or mortality rate, and finally 13 articles met the inclusion criteria and entered the meta-analysis process (Fig 1).

## Statistical analysis

In the meta-analysis, fixed and random effects models, the Cochran test, and the I2 index were utilized to analyze the data. Reviewing for publication bias with Egger's test and funnel plot,

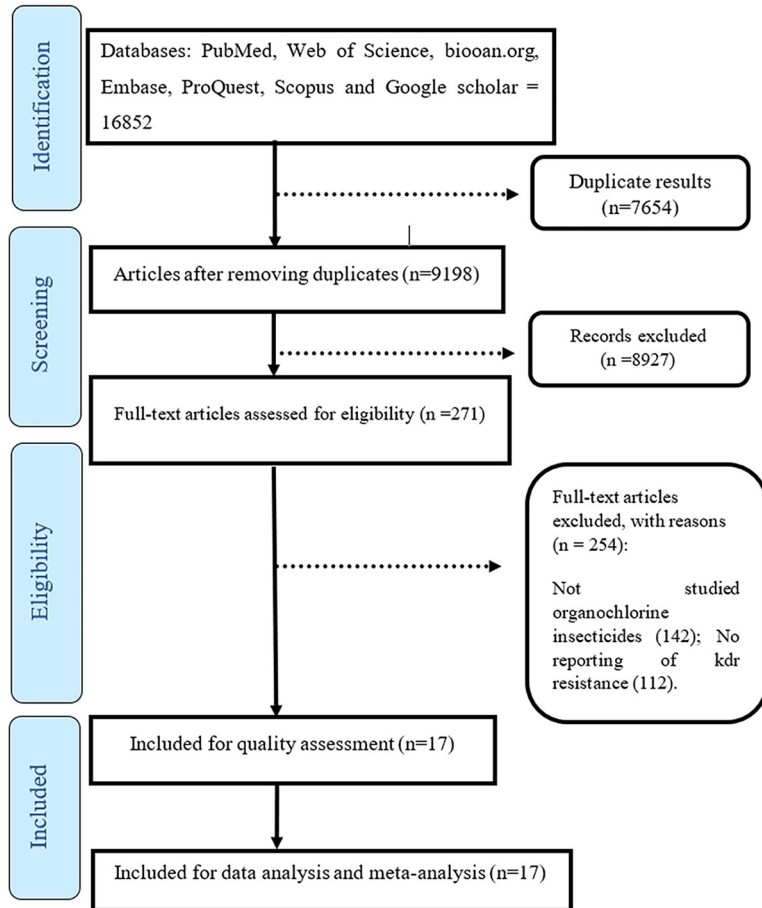

**Fig 1. PRISMA flow diagram.**

and examining the relationship between *kdr* prevalence and sample size with meta-regression. The data were analyzed utilizing version 17 of STATA.

## Results

Seventy articles with a sample size of 3733 that were conducted between 2003 and 2020 were included. 13 articles investigated the prevalence of *Kdr* resistance. Furthermore, the mortality rate against deltamethrin was investigated in 13 articles, against DDT in 11 articles, against permethrin in 11 articles, against malathion in 3 articles, and against lambda-cyhalothrin in 3 articles. The characteristics of reviewed articles are presented in Table 1.

The prevalence of *Kdr* resistance against organochlorine insecticides showed that 63.1% of *Cx. quinquefasciatus* had *Kdr* resistance (Fig 2). Thus, based on the type of resistance, the prevalence of homozygote resistance was estimated at 28.6%, and the prevalence of heterozygote resistance at 37.4% (Figs 3 and 4).

The meta-analysis of mortality rate based on the type of insecticide used to fight *Cx. quinquefasciatus* showed that the mortality rate was 46% against deltamethrin, 18.5% against DDT, 42.6% against permethrin, and 54.4% against malathion, and 53% against lambda-cyhalothrin, it was estimated. This shows that *Cx. quinquefasciatus* was most sensitive to Malathion insecticide and least sensitive to DDT (Figs 5–9).

**Table 1. Characteristics of the articles included in the meta-analysis process.**

| Author | Year of study | Place of study | Sample size | Quality assessment |
|---|---|---|---|---|
| Sarkar M ([36]) | 2010 | India | 92 | Moderate |
| Jones CM [15] | 2012 | Pemba and Unguja Island | 229 | High |
| Rai P [37] | 2019 | India (Shivmandir) | 100 | High |
| | | India (Siliguri) | 100 | |
| | | India (Jalpaiguri town) | 100 | |
| | | India (Dhupguri) | 100 | |
| | | India (Chopra) | 100 | |
| Kudoma AA [38] | 2015 | Ghana | 24 | Moderate |
| Fagbohun IK [39] | 2019 | Nigeria | 100 | High |
| Wondji CS [40] | 2008 | Sri Lanka | 207 | High |
| McAbee RD [41] | 2003 | USA | 50 | Moderate |
| Toto JC [42] | 2011 | France | - | Moderate |
| Norris LC [4] | 2011 | Zambia | 85 | Moderate |
| Shemshadian A [43] | 2020 | Iran | 100 | High |
| Kumar K [44] | 2011 | India | - | High |
| Corbel V [45] | 2007 | West Africa (Ladji) | 100 | High |
| | | West Africa (Malanville) | 100 | |
| | | West Africa (Asecna) | 100 | |
| | | West Africa (Parakou) | 100 | |
| Chen CD [46] | 2013 | Malaysia | 120 | High |
| Dery DB [47] | 2013 | Ghana | 1223 | High |
| Yadouléton A [48] | 2015 | Benin (Banikoara) | 102 | High |
| | | Benin (Kandi) | 92 | |
| | | Benin (Natitingou) | 92 | |
| | | Benin (Houeyiho) | 98 | |
| Sarkar M [49] | 2009 | India | 18 | Moderate |
| Nchoutpouen E [14] | 2019 | Cameroon | 201 | High |

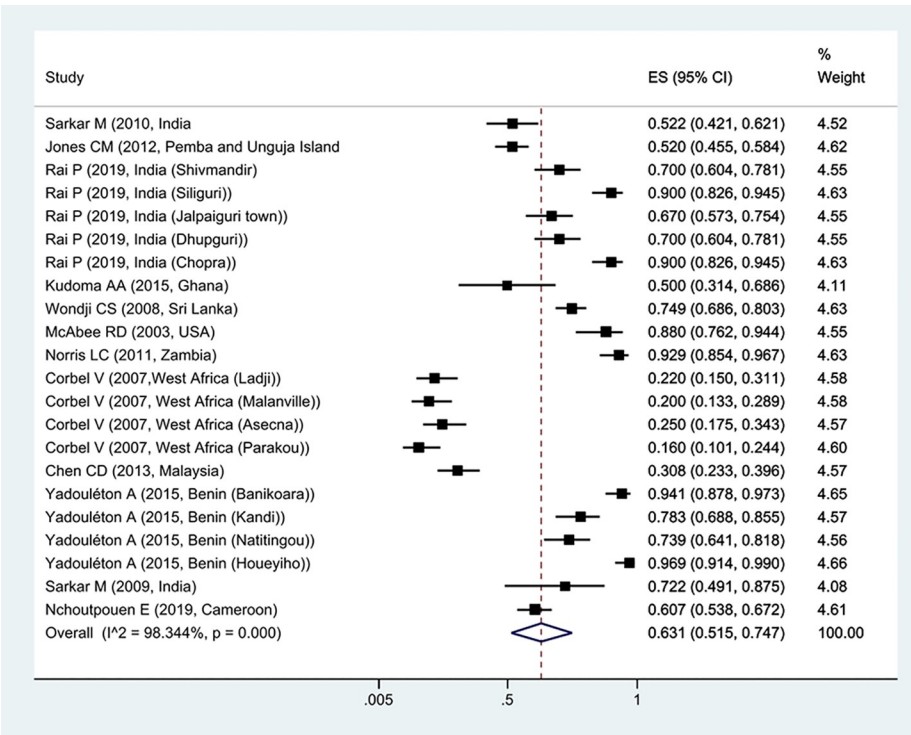

**Fig 2. Forest plots of the prevalence *Kdr* and 95% confidence interval based on the random effect model in meta-analysis.**

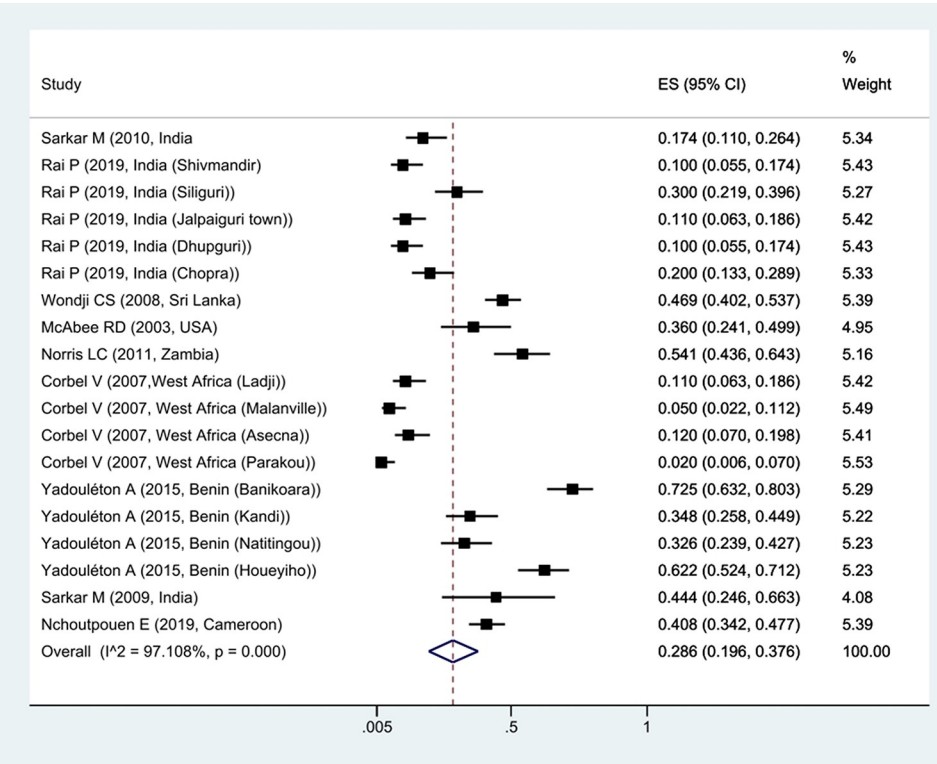

**Fig 3. Forest plots of the prevalence homozygotes resistance and 95% confidence interval based on the random effect model in meta-analysis.**

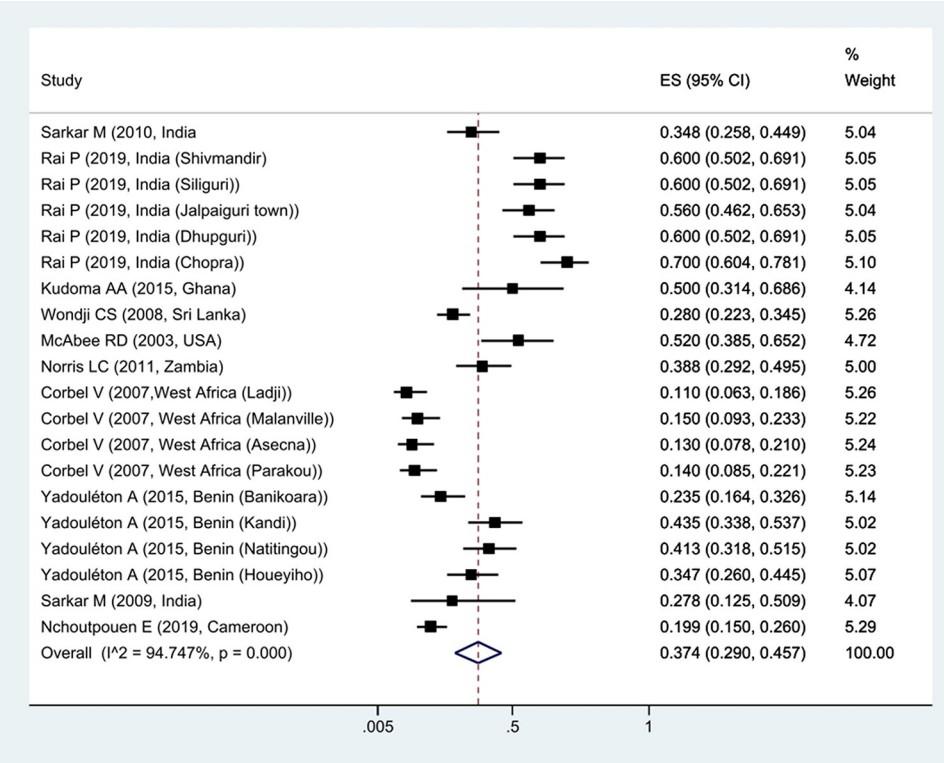

**Fig 4. Forest plots of the prevalence heterozygotes resistance and 95% confidence interval based on the random effect model in meta-analysis.**

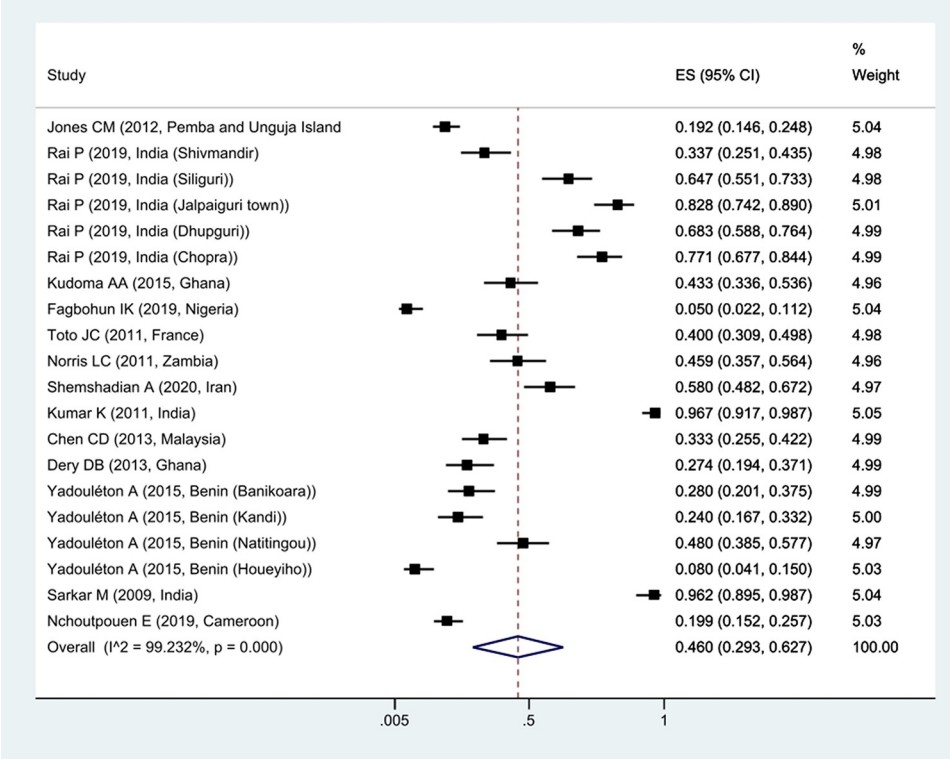

**Fig 5. Forest plots of the mortality rate *Cx. quinquefasciatus* exposed to Deltamethrin and 95% confidence interval based on the random effect model in meta-analysis.**

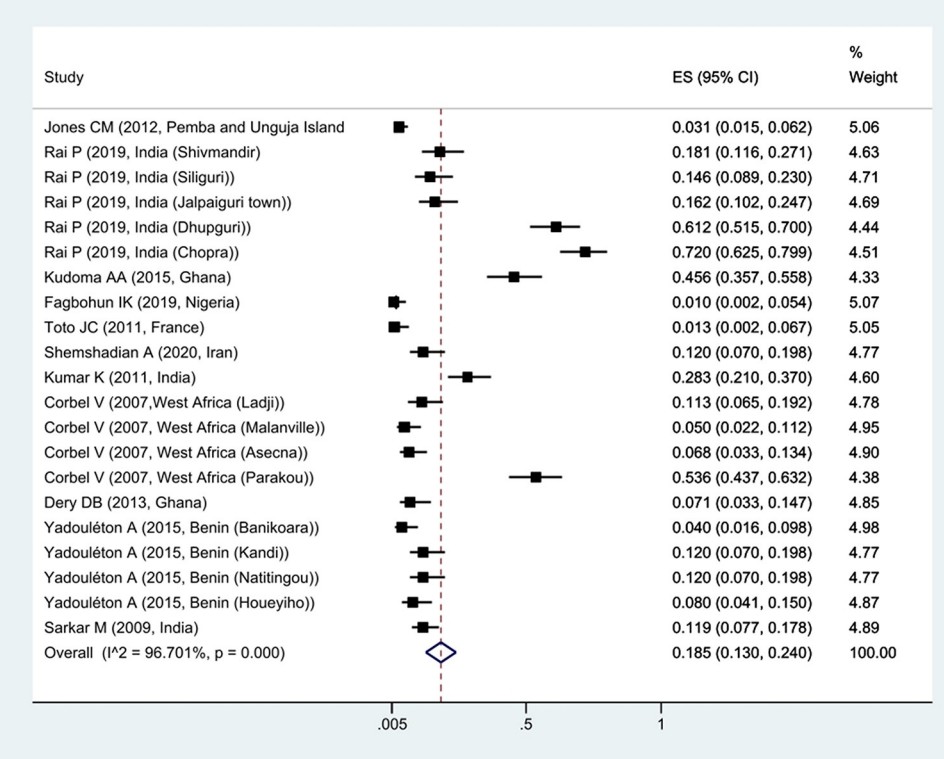

**Fig 6. Forest plots of mortality rate *Cx. quinquefasciatus* exposed to DDT and 95% confidence interval based on random effect model in meta-analysis.**

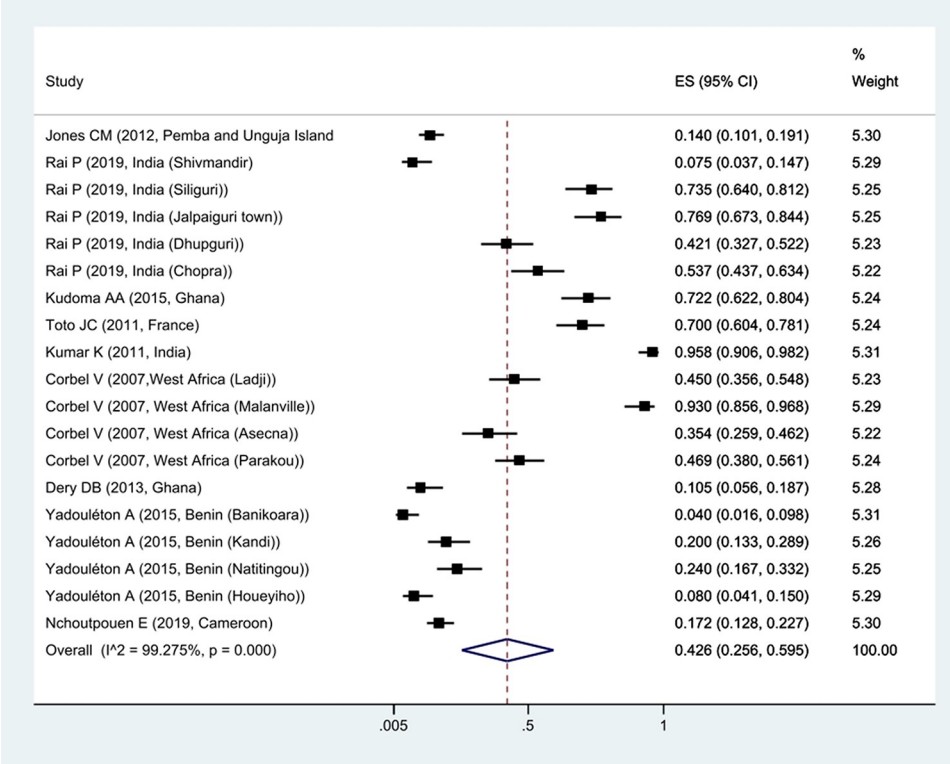

**Fig 7. Forest plots of mortality rate *Cx. quinquefasciatus* exposed to Permethrin and 95% confidence interval based on random effect model in meta-analysis.**

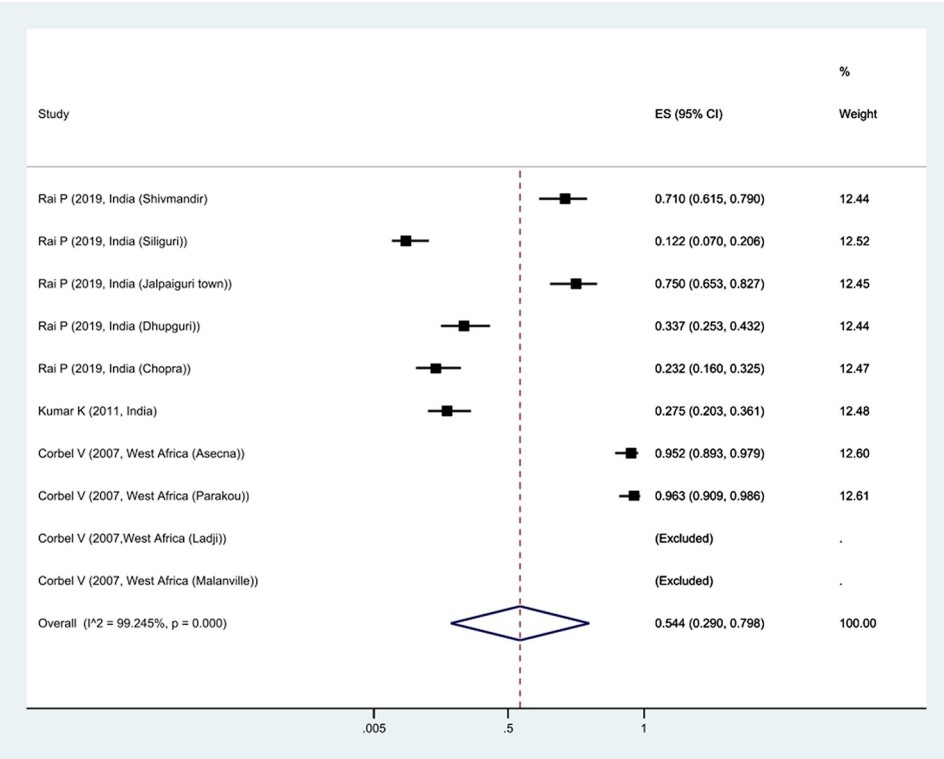

**Fig 8. Forest plots of mortality rate *Cx. quinquefasciatus* exposed to Malathion and 95% confidence interval based on random effect model in meta-analysis.**

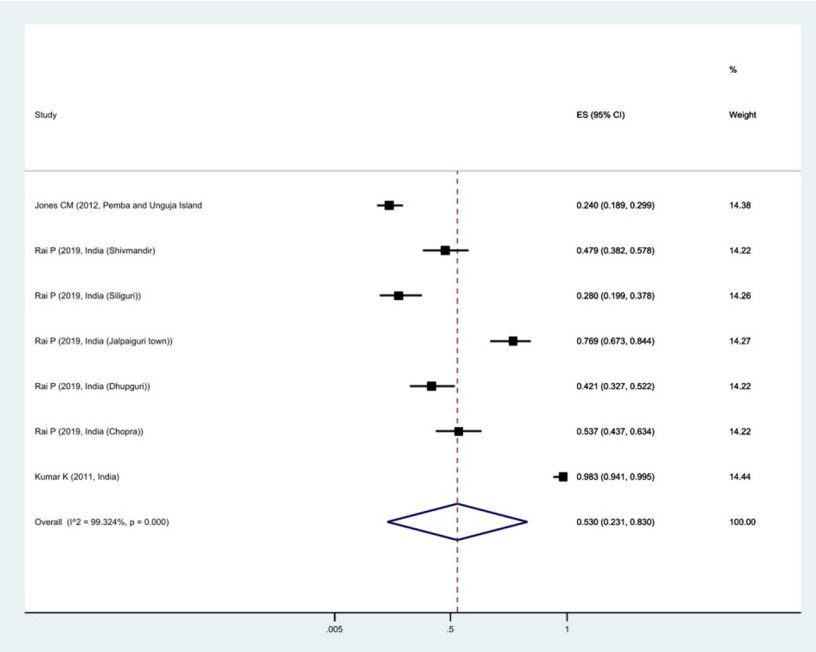

**Fig 9. Forest plots of mortality rate *Cx. quinquefasciatus* exposed to lambdacyhalothrin and 95% confidence interval based on random effect model in meta-analysis.**

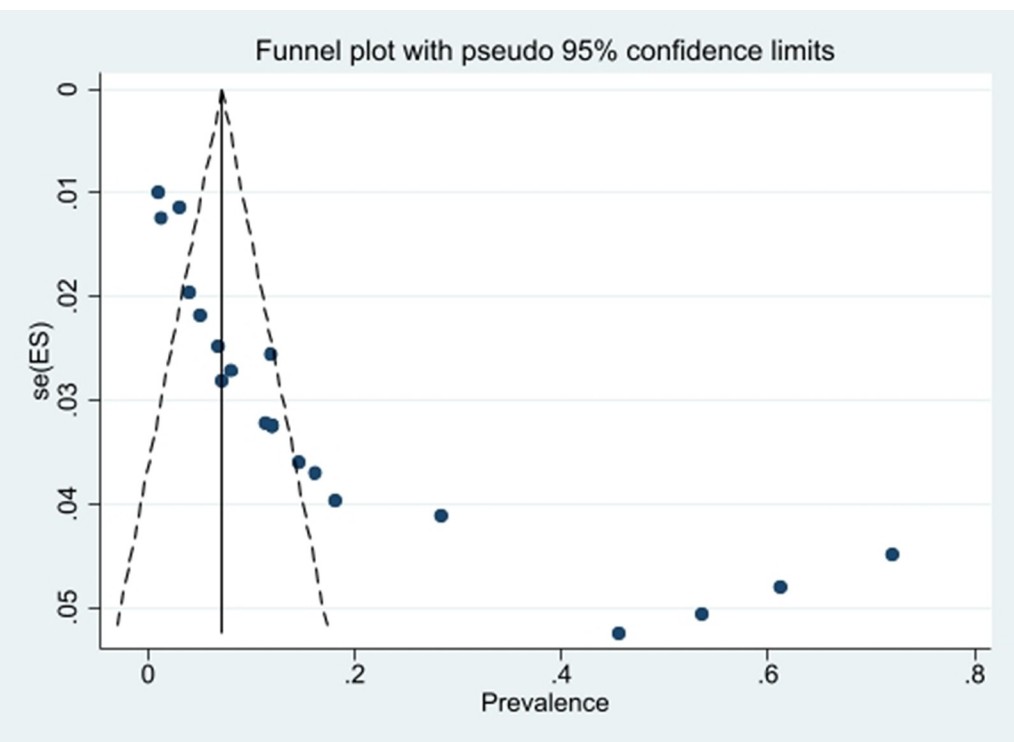

**Fig 10. Funnel plot of the mortality rate *Cx. quinquefasciatus* in the selected studies.**

In the mentioned studies, it was shown that there is multiple and cross resistance between deltamethrin, permethrin, lambdacyhalothrin, DDT and Malathion insecticides. The reason for this can be mentioned the high frequency of L1014F mutant allele and increased enzyme activity. The main mutant allele associated with kdr resistance was L1014F, which was mentioned in most studies. Among the factors related to kdr resistance were the expression of Cytochrome P450 Mono-Oxygenase, G119S mutation in the ace-1 gene, and the activity of alpha and beta esterase and glutathione S-transferase. In general, the studies showed that the kdr gene plays a role in creating cross resistance between organochlorine and pyrethroid insecticides.

Publication bias was investigated using funnel plots and Egger's test, and the relationship between sample size and mortality rate was investigated using meta-regression. Regarding the placement of studies with a high sample size below the graph, it can be mentioned that the publication bias did not occur (Fig 10). This result was confirmed due to the non-significance of Egger's test (P = 0.145). Based on the slope of meta-regression graph, with the increase in the sample size, the mortality rate decreased (Fig 11).

## Discussion

The prevalence of *Kdr* and its types in *Cx. quinquefasciatus*, as well as the mortality rate against permethrin, DDT, Malathion, deltamethrin, and lambda-cyhalothrin insecticides, were studied. Based on the results of meta-analysis, more than half of *Cx. quinquefasciatus* had *Kdr* against organochlorine insecticides. In the field of *Kdr* resistance, the prevalence of resistance heterozygotes was higher. *Kdr* mutations are the most important resistance mechanism of insects against pyrethroids and DDT [9]. The role of the *kdr* mutation in conferring resistance to DDT and permethrin insecticides on *Cx. quinquefasciatus* has been documented in

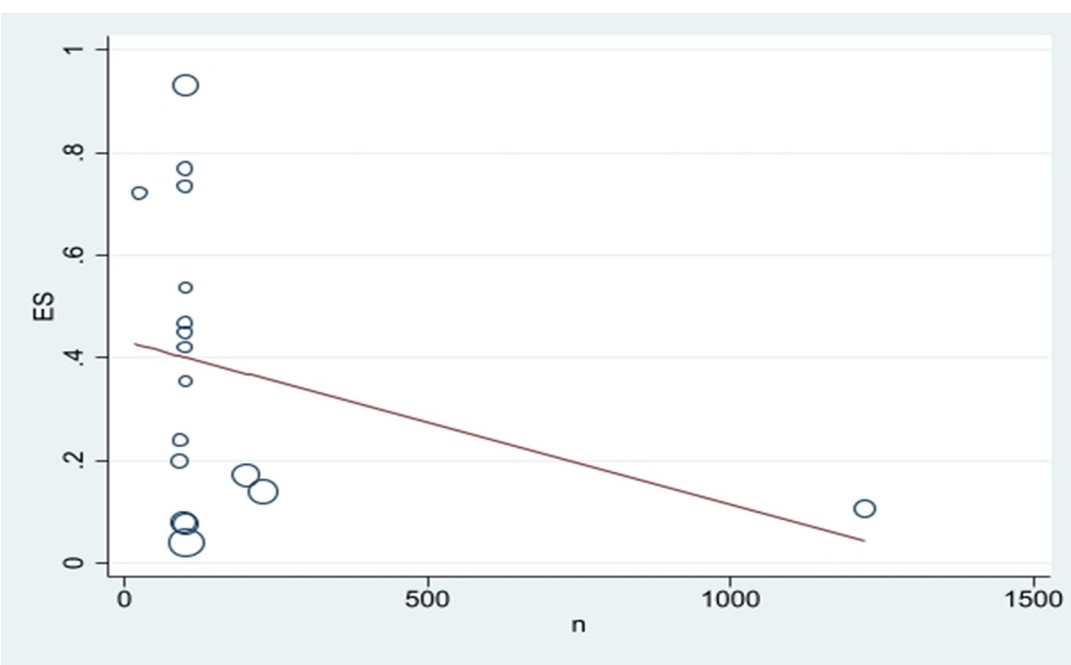

**Fig 11. Meta regression plot of the mortality rate *Cx. quinquefasciatus* of exposed to Permethrin based the study year.**

numerous studies [18–20]. The primary mechanism by which this transporter resists *Kdr* is by replacing L with F at position 1014. This substitution is involved in high resistance to DDT and pyrethroids [21,22]. In Xu et al.'s (2005) study, heterozygous and homozygous mutations for *kdr* allele in *Cx. quinquefasciatus*, it was observed that the frequency of heterozygous was much higher than homozygous [20]. In Xu et al.'s (2006) study, the relationship between *kdr* allele at the genomic DNA level, and the susceptibility and resistance of *Cx. quinquefasciatus* was not related to insecticides. Nevertheless, a robust correlation was identified between the degree of resistance and sensitivity to insecticides and the level of *kdr* allelic expression as measured by RNA allelic diversity [23]. Overall, research conducted across various global regions has yielded diverse findings regarding the prevalence of *Kdr* and its contribution to the problem of insecticide resistance. More than half of *Cx. quinquefasciatus* have *Kdr* resistance against organochlorine insecticides, and this mutation has led to high resistance against these insecticides. Considering these mutations can be exchanged among the insects, it can lead to increased resistance to other insecticides. Therefore, it is recommended before using insecticides to fight *Cx. quinquefasciatus Kdr* should be checked them and the appropriate insecticide should be used.

High resistance in *Cx. quinquefasciatus* was observed against DDT and then permethrin. In the conducted studies, it was observed that the mosquitoes resistant to DDT were also resistant to pyrethroid insecticides over time. The reason can be the indiscriminate use of pyrethroids and selection pressure, which ultimately leads to the ineffectiveness of vector control activities and pyrethroid insecticides [24,25]. Research indicated that mosquitoes that have previously exhibited resistance to DDT may develop resistance to pyrethroids as a result of cross-resistance. Particularly due to their use in IRS and LLIN, which are crucial instruments for vector control, resistance to DDT and pyrethroids has become a significant concern in the field of vector control [26,27]. In Low et al.'s (2013) study, adult mosquitoes of *Cx. quinquefasciatus* were highly resistant to permethrin and DDT. However, the larvae of *Cx. quinquefasciatus*

were relatively more resistant to malathion, and sensitive to permethrin [28]. The difference in the level of sensitivity to insecticides between larvae and adult mosquitoes is in terms of the difference in the expression of resistance gene in the larval and adult stages. Based on studies conducted on resistance to insecticides in the larval stage of *Cx. quinquefasciatus* is more common than adult mosquitoes. Excessive insecticide use and selection pressure may be the causes [29–31]. Research conducted on populations of *Cx. quinquefasciatus* has revealed a correlation between insecticide resistance in this vector species. This suggests the presence of cross-resistance, although the exact mechanism underlying this relationship remains unknown [32]. In general, cross-resistance among pyrethroids, organochlorines, organophosphorus, and carbamates was observed in vectors and it was mentioned that high levels of functional oxidases can cause cross-resistance in insecticides [33,34]. In Nazni et al.'s (2005) study, it was noted that DDT was the least effective insecticide among all insecticides tested for *Cx. quinquefasciatus* [35].

In general, the bioassay of *Cx. quinquefasciatus* is necessary to evaluate resistance to insecticides, and know the ratio of resistance to combat this vector. *Cx. quinquefasciatus* had high resistance to DDT and permethrin and relatively high sensitivity to malathion, deltamethrin, and lambdacyhalothrin. These insecticides can be used as effective insecticides to fight against it. The prevalence of resistance is different in multiple regions, and it is recommended to check the resistance before choosing insecticides for the fight.

## Conclusion

Based on the findings, more than half of *Cx. quinquefasciatus* had *Kdr*, as a result, it can be mentioned that *Kdr* mutations are the most important resistance mechanism of this vector against organochlorine insecticides. *Cx. quinquefasciatus* had relatively high resistance to DDT and permethrin insecticides and was sensitive to Malathion, deltamethrin, and lambda-cyhalothrin. This indicates that the pesticides used to combat *Cx. quinquefasciatus* are effective. To ascertain the efficacy of these pesticides, it is advised to assess their resistance first.

## Supporting information

**S1 Search Strategy. Search strategy.**
(DOCX)

## Author Contributions

**Conceptualization:** Ebrahim Abbasi.

**Data curation:** Ebrahim Abbasi, Salman Daliri.

**Formal analysis:** Ebrahim Abbasi, Salman Daliri.

**Funding acquisition:** Ebrahim Abbasi, Salman Daliri.

**Investigation:** Ebrahim Abbasi, Salman Daliri.

**Methodology:** Ebrahim Abbasi, Salman Daliri.

**Project administration:** Ebrahim Abbasi.

**Resources:** Ebrahim Abbasi.

**Software:** Ebrahim Abbasi, Salman Daliri.

**Supervision:** Ebrahim Abbasi, Salman Daliri.

**Validation:** Ebrahim Abbasi.

**Visualization:** Ebrahim Abbasi.

**Writing – original draft:** Ebrahim Abbasi, Salman Daliri.

**Writing – review & editing:** Ebrahim Abbasi.

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
