## [Decision Letter · Decision Letter 0]

14 May 2024

Dear Dr ABBASI,

Thank you very much for submitting your manuscript "Journal Article Highlight: "Knockdown Resistance (kdr) Associated Organochlorine Resistance in Mosquito-Borne Diseases (Culex quinquefasciatus): Systematic Study of Reviews and Meta-Analysis"" for consideration at PLOS Neglected Tropical Diseases. As with all papers reviewed by the journal, your manuscript was reviewed by members of the editorial board and by several independent reviewers. In light of the reviews (below this email), we would like to invite the resubmission of a significantly-revised version that takes into account the reviewers' comments. 

We cannot make any decision about publication until we have seen the revised manuscript and your response to the reviewers' comments. Your revised manuscript is also likely to be sent to reviewers for further evaluation.

Sincerely,

Mariangela Bonizzoni

Academic Editor

Paul Mireji

Section Editor

Reviewer's Responses to Questions

**Key Review Criteria Required for Acceptance?**

**Methods**

-Are the objectives of the study clearly articulated with a clear testable hypothesis stated?

-Is the study design appropriate to address the stated objectives?

-Is the population clearly described and appropriate for the hypothesis being tested?

-Is the sample size sufficient to ensure adequate power to address the hypothesis being tested?

-Were correct statistical analysis used to support conclusions?

-Are there concerns about ethical or regulatory requirements being met?

Reviewer #1: The methods have been clearly articulated.

**Results**

-Does the analysis presented match the analysis plan?

-Are the results clearly and completely presented?

-Are the figures (Tables, Images) of sufficient quality for clarity?

Reviewer #1: 1. Line 140-148: The results simply list the mortality rates of Anopheles mosquitoes to various insecticides without indicating whether there is cross-resistance between different insecticides.

2. Line 150-154: Do Culex quinquefasciatus mosquitoes with different mortality rates to different insecticides have different kdr mutations? Do kdr mutations at different loci enhance or reduce insecticide resistance? The origin and evolutionary direction of kdr mutations are the main focus of analysis.

**Conclusions**

-Are the conclusions supported by the data presented?

-Are the limitations of analysis clearly described?

-Do the authors discuss how these data can be helpful to advance our understanding of the topic under study?

-Is public health relevance addressed?

Reviewer #1: The discussion is merely a compilation of literature. For instance, the results do not indicate that kdr mutations are the most important resistance mechanism of insects against DDT, an o organochlorine There is a lack of in-depth analysis.

**Editorial and Data Presentation Modifications?**

Reviewer #1: (No Response)

**Summary and General Comments**

Reviewer #1: The novelty of the revised manuscript is average, and it does not explain why this topic was chosen. The results are overly simplistic, lacking the in-depth analysis that readers expect.

1. kdr should be italicized

2. It is well known that pyrethroids are currently the most commonly used insecticides. Why are you researching organochlorine insecticides? You need to elaborate on this.

3. Organochlorine resistance is often associated with metabolic resistance, while pyrethroids are often linked to kdr. Why have you chosen to study Knockdown resistance (kdr) Associated organochlorine Resistance? What is the deeper logic behind this?"

4. Line 140-148: The results simply list the mortality rates of Anopheles mosquitoes to various insecticides without indicating whether there is cross-resistance between different insecticides.

5. Line 150-154: Do Culex quinquefasciatus mosquitoes with different mortality rates to different insecticides have different kdr mutations? Do kdr mutations at different loci enhance or reduce insecticide resistance? The origin and evolutionary direction of kdr mutations are the main focus of analysis.

6. The discussion is merely a compilation of literature. For instance, the results do not indicate that kdr mutations are the most important resistance mechanism of insects against DDT, an o organochlorine There is a lack of in-depth analysis.

PLOS authors have the option to publish the peer review history of their article (what does this mean?). If published, this will include your full peer review and any attached files.

Reviewer #1: No
---

## [Decision Letter · Decision Letter 1]

22 Jul 2024

Dear Dr ABBASI,

We are pleased to inform you that your manuscript 'Knockdown resistance (kdr) Associated organochlorine Resistance in mosquito-borne diseases (Culex quinquefasciatus): Systematic study of reviews and meta-analysis' has been provisionally accepted for publication in PLOS Neglected Tropical Diseases.

Best regards,

Mariangela Bonizzoni

Academic Editor

Paul Mireji

Section Editor

Reviewer's Responses to Questions

**Key Review Criteria Required for Acceptance?**

**Methods**

-Are the objectives of the study clearly articulated with a clear testable hypothesis stated?

-Is the study design appropriate to address the stated objectives?

-Is the population clearly described and appropriate for the hypothesis being tested?

-Is the sample size sufficient to ensure adequate power to address the hypothesis being tested?

-Were correct statistical analysis used to support conclusions?

-Are there concerns about ethical or regulatory requirements being met?

Reviewer #1: (No Response)

**Results**

-Does the analysis presented match the analysis plan?

-Are the results clearly and completely presented?

-Are the figures (Tables, Images) of sufficient quality for clarity?

Reviewer #1: (No Response)

**Conclusions**

-Are the conclusions supported by the data presented?

-Are the limitations of analysis clearly described?

-Do the authors discuss how these data can be helpful to advance our understanding of the topic under study?

-Is public health relevance addressed?

Reviewer #1: (No Response)

**Editorial and Data Presentation Modifications?**

Reviewer #1: (No Response)

**Summary and General Comments**

Reviewer #1: (No Response)

PLOS authors have the option to publish the peer review history of their article (what does this mean?). If published, this will include your full peer review and any attached files.

Reviewer #1: No

---

## [Editor Report · Acceptance letter]

7 Aug 2024

Dear Dr ABBASI,

We are delighted to inform you that your manuscript, "Knockdown resistance (kdr) Associated organochlorine Resistance in mosquito-borne diseases (Culex quinquefasciatus): Systematic study of reviews and meta-analysis," has been formally accepted for publication in PLOS Neglected Tropical Diseases.

Best regards,

Shaden Kamhawi

co-Editor-in-Chief

Paul Brindley

co-Editor-in-Chief
